# Carbon Emissions Estimation and Spatiotemporal Analysis of China at City Level Based on Multi-Dimensional Data and Machine Learning

Xiwen Lin [1,2], Jinji Ma [1,2,*]📍, Hao Chen [3], Fei Shen [1,2], Safura Ahmad [1,2] and Zhengqiang Li [4]📍

1  School of Geography and Tourism, Anhui Normal University, Wuhu 241003, China;
   2021011469@ahnu.edu.cn (X.L.); shenfei@ahnu.edu.cn (F.S.); safahmad@uop.edu.pk (S.A.)
2  Engineering Technology Research Center of Resources Environment and GIS, Wuhu 241003, China
3  College of Physics and Electronic Information, Anhui Normal University, Wuhu 241003, China;
   ahnuchh@ahnu.edu.cn
4  State Environmental Protection Key Laboratory of Satellite Remote Sensing, Aerospace Information Research
   Institute, Chinese Academy of Sciences, Beijing 100101, China; lizq@radi.ac.cn
*  Correspondence: jinjima@ahnu.edu.cn

**Abstract:** Carbon emissions caused by the massive consumption of energy have brought enormous pressure on the Chinese government. Accurately and rapidly characterizing the spatiotemporal characteristics of Chinese city-level carbon emissions is crucial for policy decision making. Based on multi-dimensional data, including nighttime light (NTL) data, land use (LU) data, land surface temperature (LST) data, and added-value secondary industry (AVSI) data, a deep neural network ensemble (DNNE) model was built to analyze the nonlinear relationship between multi-dimensional data and province-level carbon emission statistics (CES) data. The city-level carbon emissions data were estimated, and the spatiotemporal characteristics were analyzed. As compared to the energy statistics released by partial cities, the results showed that the DNNE model based on multi-dimensional data could well estimate city-level carbon emissions data. In addition, according to a linear trend analysis and standard deviational ellipse (SDE) analysis of China from 2001 to 2019, we concluded that the spatiotemporal changes in carbon emissions at the city level were in accordance with the development of China's economy. Furthermore, the results can provide a useful reference for the scientific formulation, implementation, and evaluation of carbon emissions reduction policies.

**Keywords:** carbon emissions; machine learning; multi-dimensional data; spatiotemporal analysis; China

## 1. Introduction

Human activities consume energy and produce a large amount of carbon dioxide, which aggravates the greenhouse effect and leads to increasing global temperature. Human society and the natural environment are facing significant threats [1]. Therefore, in countries around the world, an increasing number of people are becoming concerned about energy conservation and reductions in carbon emissions [2,3]. As one of the world's largest carbon emitters [4], China has made a solemn commitment to the world to reach the carbon emission peak by 2030 and be carbon neutral by 2060 [5,6]. Admittedly, it is a great challenge for China to achieve this goal. Under the guidance of the concept of "green", "low carbon", and "sustainable" socio-economic development, studying the spatiotemporal dynamics of China's carbon emissions is vital for decision makers to attain the goals.

Considering the lack of official city-level energy consumption statistics, refined scales of carbon emissions estimations are urgently needed. Recently, several methods to estimate carbon emissions have emerged, which can be divided into two groups. The first group utilizes the collection of various fossil energy consumptions directly from statistical

yearbooks at the province level, which was proposed by the Intergovernmental Panel on Climate Change (IPCC). The total consumption of various fossil fuels was multiplied by their average low calorific value and carbon emission coefficient to obtain the carbon emissions [7], then the spatiotemporal analysis and influencing factor analysis of carbon emissions at the nation level or province level were carried out [8–10]. This method was widely used [11,12], but the estimated carbon emissions were nation-level or province-level data. Due to the lack of city-level carbon emissions data, the carbon emissions reduction measures proposed by the government did not work as expected.

The second group centers on applications of nighttime light (NTL) data. It is known that NTL data can well reflect human socio-economic activities, which is an effective surrogate variable for evaluating various socio-economic indicators, including electricity consumption [13–15], urbanization development [16,17], economic development [18], population density [19], and so on. Human carbon emissions and socio-economic activities are closely related. Therefore, NTL data can be used to estimate carbon emissions accurately [20–22]. In 1997, Elvidge et al. [23] first demonstrated the logarithmic relationship between NTL data and greenhouse gases at the nation level. This provided a theoretical basis for scholars to use NTL data to estimate the carbon emissions of energy consumption at a refined scale. In recent years, plenty of works have been done on carbon emission estimation at various scales. Shi et al. [24] combined the Defense Meteorological Satellite Program's Operational Linescan System (DMSP-OLS) data with carbon emissions data to simulate the spatiotemporal carbon emission in China. Similarly, Zhong et al. [25] adopted a multi-period mask denoising method that combined the National Polar-Orbiting Partnership Satellite's Visible Infrared Imaging Radiometer Suite (NPP-VIIRS) in "The Belt and Road" region with a spatial resolution of 0.5 km $\times$ 0.5 km to analyze the spatiotemporal characteristics of carbon emissions in this region. Lv et al. [26] integrated DMSP-OLS and NPP-VIIRS data, and established a comprehensive dataset of nighttime lights from 1992 to 2016 to build a carbon emission estimation model that can analyze the spatiotemporal analysis of carbon emissions at multiple scales.

However, most of the studies in the second group [24–26] only considered the linear relationship between NTL data and carbon emissions statistics (CES) data. In fact, the relationship between NTL data and carbon emissions is not simply linear. Carbon emissions are also related to urban expansion, industrial structure, and energy efficiency [27,28]. Estimating carbon emissions based only on NTL data may be inaccurate. Therefore, this method proposed using multi-dimensional data, including land use (LU), land surface temperature (LST), statistical value of the added value of secondary industry (AVSI) by region, and the NTL. On the other hand, as a new star in interdisciplinary science, machine learning has been successfully applied in many fields, such as image segmentation [29] and data mining [30,31]. However, exploring multi-dimensional data fusion to estimate carbon emissions is also a challenge for the machine learning method [32].

This study aimed to estimate city-level carbon emissions by using multi-dimensional data and machine learning, which can overcome the shortcomings of only using NTL data. The contributions of this study are as follows: (1) Machine-learning models were built to train carbon emission estimation models at the province level by using multi-dimensional data; (2) to deal with the lack of city-level carbon emission statistics in the statistical yearbook, the city-level multi-dimensional data input model was used to estimate the city-level carbon emissions from 2001 to 2019; and (3) by using the methods of linear trend analysis and standard deviation ellipse analysis, the spatiotemporal changes in carbon emissions at the city level in China were analyzed.

## 2. Materials and Methods

Figure 1 shows the flowchart of this work with the following steps. Firstly, the NTL, LU, LST, AVSI, and CES data were preprocessed according to the respective steps given in Section 2.2 to ensure that all training data were at the province level. Secondly, three models, including the multiple linear regression (MLR) model, random forest (RF) model, and deep

neural network ensemble (DNNE) model, were adopted to train the data. Furthermore, the root-mean-square error (RMSE) and determination coefficient ($R^2$) were utilized to evaluate the performances of the models. The city-level data were processed using the same preprocessing method, followed by city-level carbon emissions estimation with the help of the optimal model. In the end, based on the long-time series datasets of city-level carbon emissions, the spatiotemporal characteristics were analyzed.

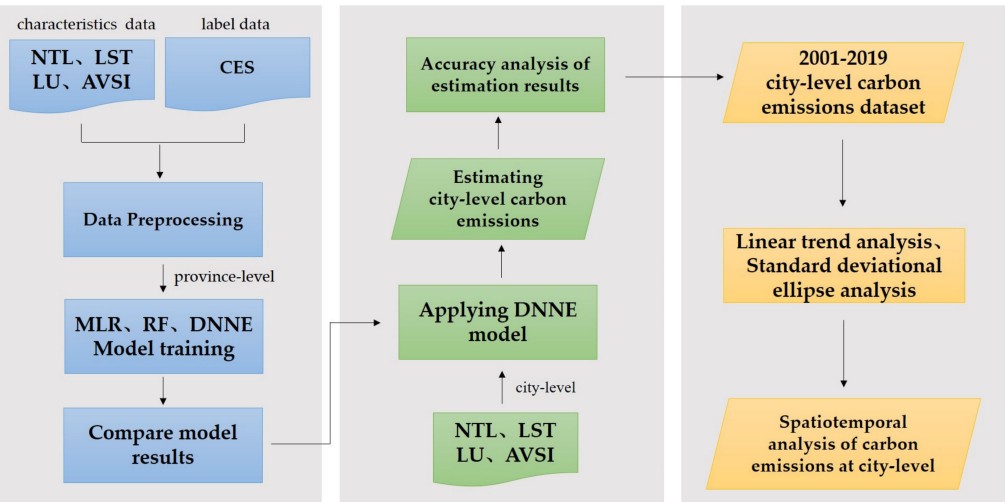

**Figure 1.** Flowchart of the study.

*2.1. Study Area*

The study area covered most of mainland China. Due to the lack of energy statistics data, Hong Kong, Macau, Taiwan, and Tibet were excluded. There were significant differences in regional energy consumption due to unbalanced economic development in the country. As a result, the entire experimental area was divided into three regions according to the regional division method of the Resource and Environmental Science and Data Center (https://www.resdc.cn (accessed on 15 July 2021)); these included the eastern, central, and western regions (Figure 2).

*2.2. Data Source and Preprocessing*

Multi-dimensional data were used to improve the machine-learning estimation model of carbon emissions. The characteristics data included NTL data, LU data, LST data, and AVSI data of the region. The label data consisted of emissions from 17 types of fossil fuels burned in 47 socio-economic sectors and emissions from the cement production industry. These were derived from the province-level CES dataset published by Shan et al. [33]. However, the statistics of Hainan province in 2002 were missing. As a supplement, this study utilized the interpolation method to estimate the carbon dioxide emissions of Hainan Province in 2002. Figure 3 shows summarized carbon emission statistics of the province from 2001 to 2019. Table 1 outlines all the data sources and their descriptions. As the data could not be used directly as the input of the machine-learning model, it was necessary to process it and extract characteristics to enable the model to learn better and improve its performance.

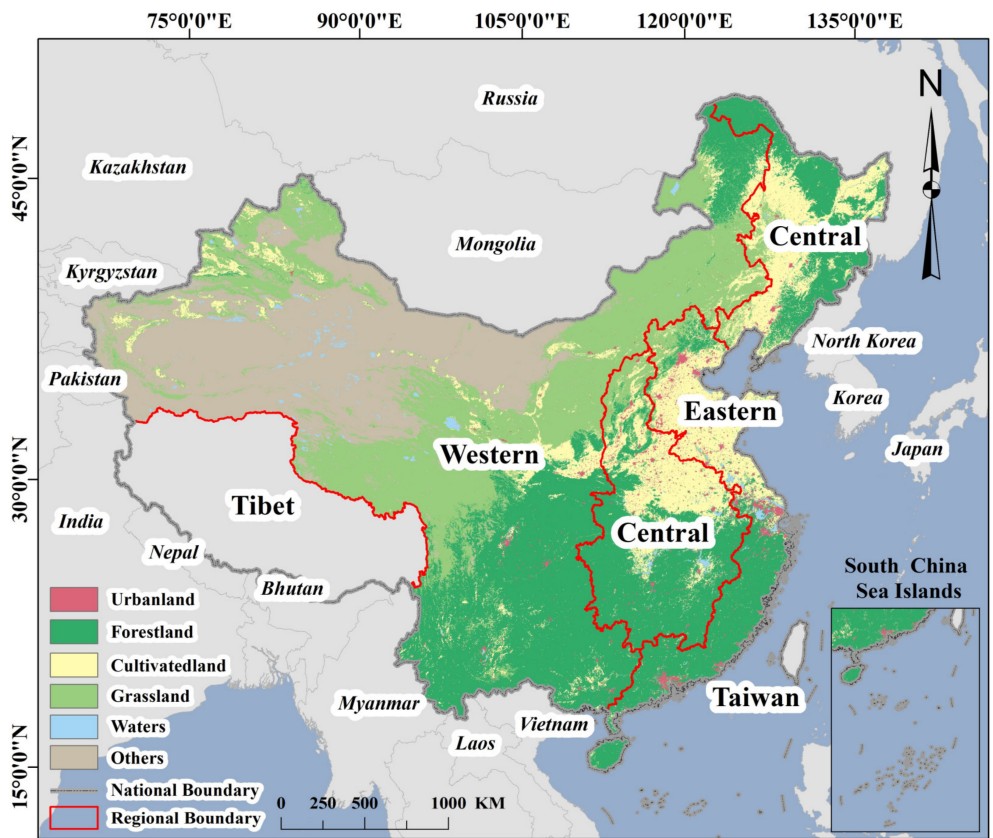

**Figure 2.** Regional division.

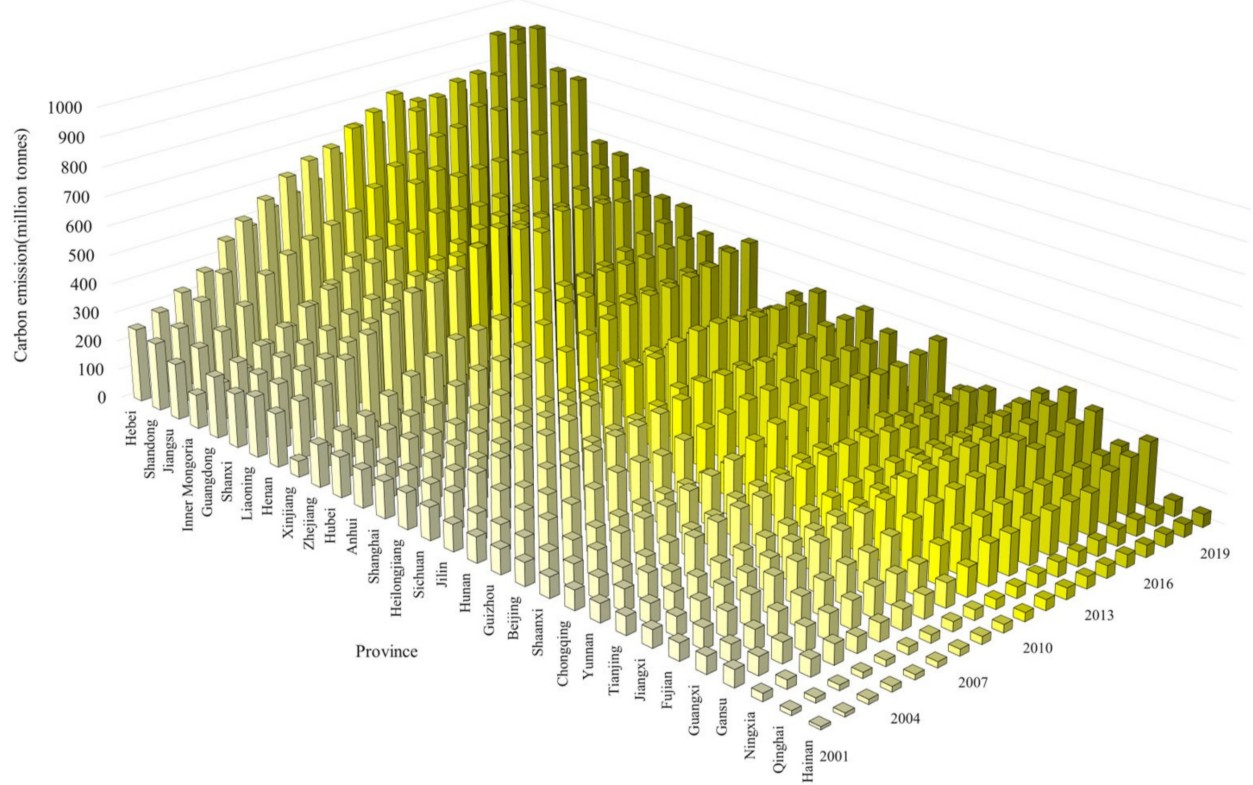

**Figure 3.** Statistics on carbon emissions by province.

**Table 1.** Data list.

| Data | Data Description | Time Interval | Source |
|---|---|---|---|
| DMSP-OLS | Annual DMSP-OLS nighttime stable light data with a spatial resolution of 1 km × 1 km | 2001–2013 | NOAA (https://ngdc.noaa.gov/eog/dmsp/downloadV4composites.html (accessed on 26 January 2020)) |
| NPP-VIIRS | Monthly NPP-VIIRS nighttime light data with a spatial resolution of 0.75 km × 0.75 km | 2012–2019 | EOG (https://eogdata.mines.edu/nighttime_light/monthly/v10/ (accessed on 26 January 2020)) |
| Land use | Derived using the supervised classification of MODIS Terra and Aqua reflectance data at a spatial resolution of 0.5 km × 0.5 km | 2001–2019 | NASA (https://lpdaac.usgs.gov/products/mcd12q1v006 (accessed on 19 January 2022)) |
| Land surface temperature | Provides 8-day average surface temperature with a resolution of 1 km × 1 km | 2001–2019 | NASA (https://lpdaac.usgs.gov/products/mod11a2v006 (accessed on 29 December 2021)) |
| Boundaries | Shapefile of province-level and city-level regions | 2015 | Resource and Environmental Science and Data Center (https://www.resdc.cn (accessed on 15 July 2021)) |
| Added value of secondary industry | The unit output value-added value of the secondary industry in a certain period | 2001–2019 | China Statistical Yearbook (http://www.stats.gov.cn (accessed on 19 December 2021)) |
| Carbon emission statistics | Total carbon emissions of energy consumption in 30 provinces in the study area | 2001–2019 | Carbon Emission Accounts and Datasets (https://www.ceads.net.cn (accessed on 19 October 2021)) |

### 2.2.1. Land Use

In this study, the MCD12Q1v006 annual products from 2001 to 2019 were used to extract the characteristics of urban land types. The LU data were reclassified into six land use types: urban land, forest land, cultivated land, waters, others, and grassland, then resampled to a 1 km ×1 km spatial resolution. Many studies have shown that there is a close relationship between land use types and carbon emissions [34,35], among which urban land carries the vast majority of human social and economic activities. Therefore, urban land is the main source of carbon emissions [36]. This study used the provincial administrative boundary data mask to count the urban land area, which was then used as the annual LU data characteristic of each province.

### 2.2.2. Land Surface Temperature

Urban development is accompanied by energy consumption. Studies have shown that artificial heating, buildings, and roads inside the city can make the LST significantly higher than outside the city. Thus, LST is related to the use of energy consumption [37]. The LST product selected in this study was MOD11A2v006, which is an 8-day synthetic diurnal surface temperature product. Initially, the product was averaged and synthesized in units of years with the help of Google Earth Engine. After that, the average value of pixels within each province was calculated through the provincial administrative boundary data to obtain the annual average LST data characteristic of each province.

### 2.2.3. Nighttime Light

NTL data is a crucial indicator of human activities. It is also ideal data to evaluate the intensity of energy consumption. The NTL data used in this study was published in our previous work [38], and was acquired from two satellite sensors, including the DMSP-OLS (satellite time interval from 1992 to 2013) and NPP-VIIRS (satellite time interval from 2012 to 2021). Most scholars only adopted one of these sensors to estimate carbon

emissions, which is challenging when conducting a long-term comprehensive dynamic analysis of carbon emissions [24,25]. Taking the Sicily area of the F12 satellite in 1999 as the light-invariant area, the DMSP-OLS data of China from 1992 to 2013 was corrected based on the statistical relationship of the satellite data of other years through the quadratic polynomial function [39]. In addition, the DMSP-OLS was obtained from four satellites (F14 (1997–2003), F15 (2000–2007), F16 (2004–2009), and F18 (2010–2013)), which were merged within the year as shown in Equation (1) to ensure the continuity of the digital numbers (DNs) [26].

$$DN_{(y,i)} = \frac{DN_{(y,i)}^{a} + DN_{(y,i)}^{b}}{2} \tag{1}$$

where $y$ = 2001, 2002, ... 2007; and $DN_{(y,i)}$ denotes the $i$ pixel DN value in $y$ year from $a$ satellite and $b$ satellite.

On the other hand, after being synthesized by annual products, resampled to the spatial resolution of 1 km $\times$ 1 km, denoised, corrected by the Biphasic Dose Response model, and smoothed by Gaussian lowpass filtering to obtain the BDRVIIRS products, the NPP-VIIRS data can be consistent with DMSP-OLS in spatial resolution. Figure 4 shows the changing trend of the sum of DN values in China after mutual calibration, in which it can be seen that the sum of DN values has been increasing with the development of China's economy. This study calculated the sum of DN values in each province through the administrative boundary, which was used as the annual NTL data characteristic of each province.

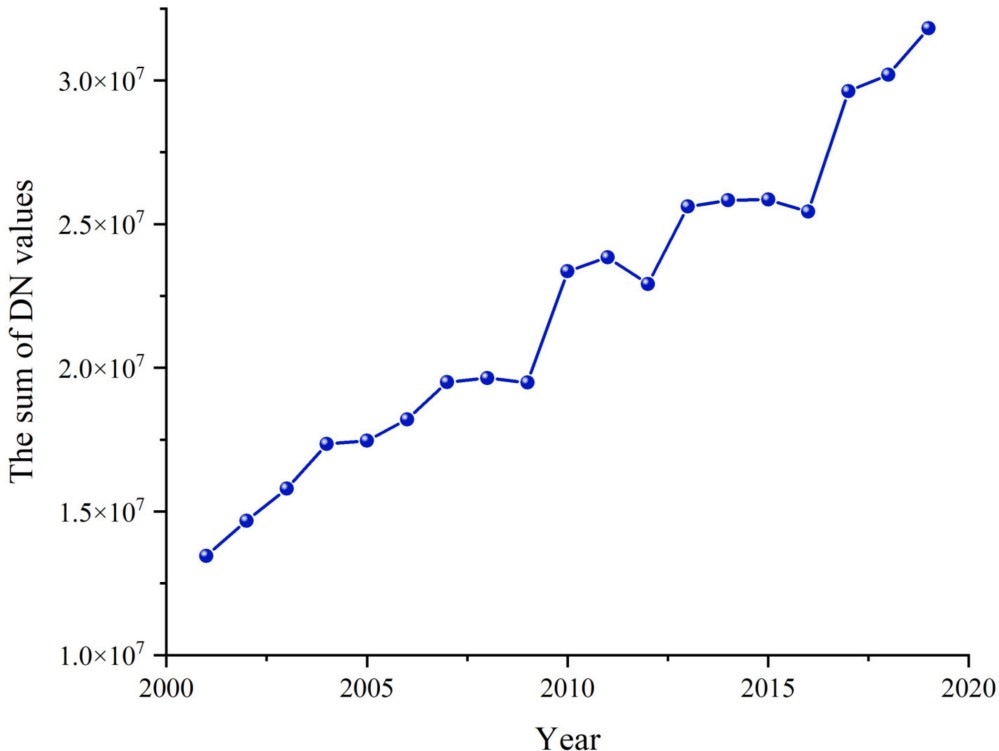

**Figure 4.** Sum of DN values in China from 2001 to 2019.

### 2.2.4. Added-Value Secondary Industry

Carbon emissions are mainly due to industrial consumption [33]. In this regard, AVSI data can be used to measure the industrial development status of a region [40]. Therefore, this research adopted the AVSI data characteristics of each province in the study area that were published in the *China Statistical Yearbook* from 2001 to 2019.

### 2.3. Machine-Learning Algorithms

Before model training, the training set and the test set were randomly divided in a ratio of 3:1 (divided into three regions for training, as noted in Section 2.1). The Yeo–Johnson standardization method for nonlinear transformation was adopted, which can effectively alleviate the problem of data skew and make the numerical interval of all features more accurate [41]. In addition, all models adopted the same test and training data to obtain comparable performance evaluation results. Moreover, two evaluation indicators (RMSE, $R^2$) were utilized to evaluate the result. RMSE is a quantitative method for evaluating the effectiveness of regression methods. $R^2$ is called the adjusted coefficient of determination, which is an indicator to measure the quality of regression fitting. A high $R^2$ value indicates that all predictions fully correspond to the actual results.

### 2.3.1. Multiple Linear Regression

The Multiple Linear Regression (MLR) model is a parametric method in machine learning that obtains the final predicted value by assigning corresponding weights to different variables for performing weighted summation [42]. Equation (2) of the multiple linear regression model of the dependent variable *Y* (carbon emissions) and the independent variable *X* (multi-dimensional data) can be calculated as follows:

$$Y = XW + b + \varepsilon \tag{2}$$

where *Y* denotes the column vector composed of the predicted carbon emission values of each sample; *X* denotes a matrix composed of multi-dimensional data features; *W* denotes the weight vector corresponding to the multi-dimensional data; *b* denotes the bias vector, in which all element values are equal; and $\varepsilon$ denotes that the model cannot be fitted to the combined residual part. The weight *W* and bias *b* are learnable parameters that can be obtained by algorithms such as closed-form solution and gradient descent.

### 2.3.2. Random Forest

Random Forest (RF) is a nonlinear modeling tool proposed by Breiman in 2001. The basic unit of RF is a binary decision tree, which can simulate the complex relationship between dependent variables and independent variables [43]. It uses the bagging method to integrate the binary decision tree. The process of its construction includes sampling with replacement, which means that the construction process of any two trees does not affect each other. Therefore, the RF model can easily realize the parallel construction of the binary decision tree. In the regression scenario, after completing the construction of several binary decision trees, the predicted values are averaged to obtain the final integrated prediction result [44,45]. RF has many hyperparameters, which can significantly affect the fitting ability and generalization ability of the random forest model. In this study, the optimal number of binary decision trees, the optimal maximum number of features, and the optimal maximum tree depth were the three primary hyperparameters used for predictors, and through a sensitivity analysis and grid search continuous adjustment, the parameter combination that optimized the generalization ability of the model was finally selected.

### 2.3.3. Deep Neural Network Ensemble

The deep neural network ensemble (DNNE) is a widely used deep learning model to handle various classification and regression problems [46,47]. Its basic unit is the neuron, and each neuron consists of a linear function and an activation function. The linear function performs a weighted summation of the input data from all neurons in the previous layer for sending it to the activation function to perform the nonlinear transformation, then the transformation result is output to the neurons of the next layer as input. A single neuron receives the output of all neurons in the previous layer as its input at the same time; after the operation, it outputs to all the neurons in the next layer as input. The neurons in the same layer are parallel and do not affect each other, and they are also connected, thus

forming a neural network model. When a neural network model has many layers, it reaches a sufficiently deep level, which is called a deep neural network model [48,49]. The weights of the initial model are randomly initialized, and the training of the model is performed alternately through forwarding propagation and backpropagation. Through continuous iteration, the cost function is gradually minimized until the model converges, and finally, the optimal parameters are trained.

Since this research belonged to a regression task, the cost function was set to the mean-square error, and the activation function of each layer was set to rectify the linear unit [50]. To improve training efficiency, the Adam optimization algorithm was adopted instead of the standard gradient descent algorithm [51]. It took 0.001 as the initial learning rate, then adaptively changed the learning rate. During the training process, if the cost function of the test set had not decreased 15 times, the learning rate was reduced to 95% of the original. In addition, the entire training set had 10,000 rounds of iteration. During the training process, parameters were saved at the end of each round, and finally the parameters corresponding to the lowest cost function of the test set were determined as the final parameters of the model.

*2.4. Analysis of the Spatiotemporal Characteristics of Carbon Emissions*

2.4.1. Linear Trend Analysis

Linear trend analysis is a widely used spatiotemporal analysis method [26,52]. The changes in the slope of China's city-level carbon emissions in the three time periods of 2001–2006, 2006–2012, and 2012–2019 were obtained by establishing a linear regression model of energy consumption carbon emissions and time. The slope can be calculated by using Equation (3) as follows:

$$Slope = \frac{n \times \sum_{i=1}^{n} y_i C_i - \sum_{i=1}^{n} y_i \times \sum_{i=1}^{n} C_i}{n \times \sum_{i=1}^{n} y_i^2 - \left( \sum_{i=1}^{n} y_i \right)^2} \tag{3}$$

where *Slope* denotes inter-annual slope changes in city-level carbon emissions, $n$ denotes the time sequence (2001–2019), $y_i$ denotes the $i$ year (2001 is the first year), and $C_i$ denotes the carbon emissions in $i$ year. Table 2 is the development of the classification standards for carbon emission trends.

**Table 2.** Change trend classification.

| Growth Type | Rapid Negative Growth | Relatively Rapid Negative Growth | Slow Growth |
|---|---|---|---|
| Slope | <k − s | K − s ~ 0 | 0 ~ k + 0.5 s |
| Growth type | Relatively slow growth | Relatively fast growth | Rapid growth |
| Slope | k + 0.5 s ~ k + s | k + s ~ k + 2 s | >k + 2 s |

Note: k is the mean slope, and s is the standard deviation of the slope.

2.4.2. Standard Deviational Ellipse

To further explore the directionality and spatiotemporal characteristics of carbon emissions, the standard deviational ellipse (SDE) was applied to express the changing characteristics of the geographic and spatial distribution of carbon emissions at the city level in China. The SDE is a classical algorithm that reveals the distribution characteristics and evolution process of a set of discrete point sets [53]. Other characteristics such as the ellipse center, long-axis, short-axis, and azimuth can be obtained by calculation. Consequently, by plotting the SDE of China's regional carbon emissions from 2001 to 2019 and comparing the evolution characteristics of these ellipses over time, the spatiotemporal characteristics of China's carbon emissions could be visually revealed.

## 3. Results and Discussion

### 3.1. Model Selection and Application

3.1.1. Model Comparison

Three models, including MLR, RF, and DNNE, were adopted for training in this study. Simultaneously, multi-dimensional data were used; i.e., provincial LST, LU, AVSI, NTL, and CES data. Table 3 describes the performance of the three models using the training and test sets. Model training was carried out on the Google Colaboratory cloud platform. MLR is a simple model, so the training time was short, while RF and DNNE are more complex models, and their training times were similar and longer. This showed that the $R^2$ of the DNNE model was larger than that of the MLR model and RF model in the study areas, indicating the estimations and the actual results corresponded well. In addition, the RMSE of the DNNE model was smaller than other two methods, indicating the accuracy of estimations was better.

**Table 3.** Comparison of model results.

| Study Area | Methods | Training Set | | Test Set | | Time |
|---|---|---|---|---|---|---|
| | | $R^2$ | RMSE | $R^2$ | RMSE | |
| Eastern | MLR | 0.8549 | 91.1901 | 0.8319 | 106.5336 | 5 S |
| | RF | 0.9646 | 46.3780 | 0.9356 | 55.2019 | 438 S |
| | DNNE | 0.9959 | 14.9305 | 0.9899 | 22.6544 | 445 S |
| Central | MLR | 0.7231 | 65.4376 | 0.5863 | 79.6208 | 5 S |
| | RF | 0.9390 | 31.0839 | 0.8936 | 40.3151 | 423 S |
| | DNNE | 0.9901 | 11.1858 | 0.9901 | 12.2030 | 424 S |
| Western | MLR | 0.6344 | 81.6179 | 0.5284 | 96.3817 | 5 S |
| | RF | 0.8869 | 45.1309 | 0.7957 | 56.8278 | 430 S |
| | DNNE | 0.9939 | 10.6804 | 0.9786 | 13.8629 | 433 S |

In the ranking of the importance of image factors in Table 4, the LU and the AVSI in the eastern, central, and western regions were all important factors. This was mainly due to the massive energy consumption caused by urban expansion and industrial production, resulting in massive carbon emissions [54]. This work combined the province-level training and test sets in each region, and reinput them into the trained DNNE model to estimate the carbon emissions of each province. Figure 5 depicts the scatter plot and histogram of the statistical carbon emissions value and the estimated value of the DNNE model at the province level of the eastern, central, and western regions, respectively. The results showed that the $R^2$ of the scatter plots in the three regions were all greater than 0.99, the regression line slope was close to 1, and RMSE was less than 20. Therefore, we concluded that the DNNE model was the best method to establish the regression relationship between the multi-dimensional data characteristics and carbon emissions.

**Table 4.** Ranking of the importance of influencing factors.

| | LU | AVSI | NTL | LST |
|---|---|---|---|---|
| Eastern | 0.31 | 0.32 | 0.21 | 0.16 |
| Central | 0.27 | 0.33 | 0.20 | 0.20 |
| Western | 0.34 | 0.27 | 0.27 | 0.12 |

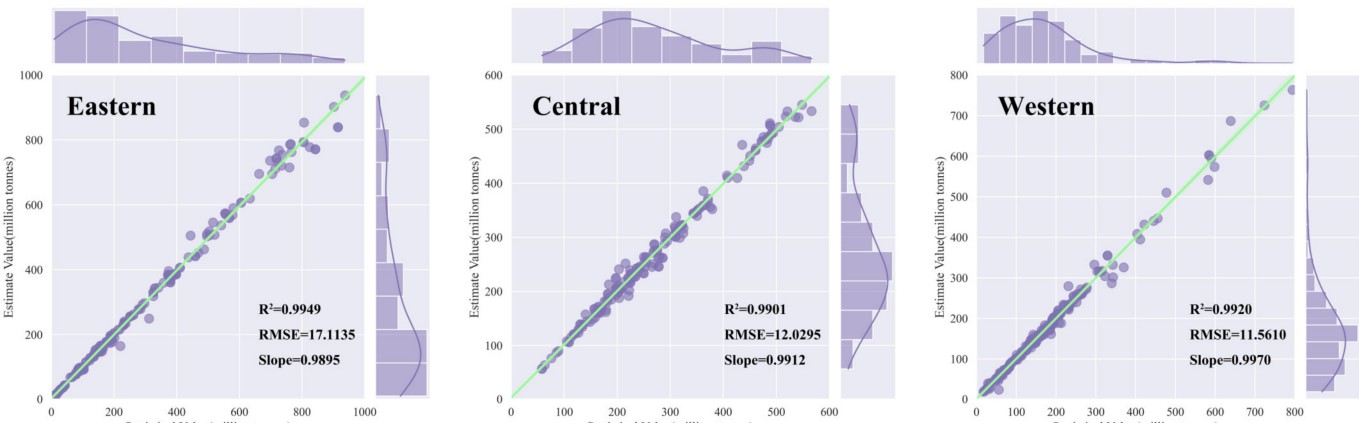

**Figure 5.** Histograms, scatter plots, and linear fitting lines of estimated and statistical values in the three regions.

### 3.1.2. Model Application and Evaluation

To estimate carbon emissions at a finer city level, it was assumed that the province-level statistical relationship between the geophysical characteristics and carbon emissions in this study was still maintained at the city level. According to the previous method of data preprocessing, based on the administrative boundaries, the city characteristics from 2001 to 2019 were extracted from NTL, LST, LU, and AVSI data. By inputting each set of characteristics into the trained DNNE model, after model inference, the carbon emissions results of cities in the experimental area from 2001 to 2019 were obtained. As shown in Figure 6, China cities with high energy consumption and carbon emissions were primarily located in the Bohai Rim, eastern coastal urban agglomerations, and midwestern regions such as Chongqing, Wuhan, and Chengdu, which was consistent with the findings of Sun et al. [22] and LV et al. [26].

Figure 7 shows the comparison between partial city-level carbon emission statistics [55] and the estimated results of the proposed method. The cities whose estimated value is higher than the statistical value were mainly Shijiazhuang, Chengdu, Quanzhou, Wenzhou, and Xiamen. The reasons for the high value of these cities were different (Table 5). Among them, Shijiazhuang, Quanzhou, and Wenzhou had high estimations due to the high proportion of urban coal and coke consumption in total energy consumption. However, the results for Chengdu and Xiamen were due to the relatively developed tertiary industry and relatively high NTL, resulting in high estimations. The cities whose estimated value was lower than the statistical value mainly included Hangzhou, Dalian, Jinan, Zibo, Changsha, Zhenjiang, Fuxin, and Xinyu. Energy consumption per unit of GDP (ESGDP) reflects the level of energy technology in a region. Hangzhou's ESGDP was only 0.68; the city's energy technology level is relatively high, resulting in a low final estimate of urban carbon emissions. Urban land was the main source of carbon emissions [36], with low estimates for Dalian, Jinan, Zibo, Changsha, Zhenjiang, Fuxin, and Xinyu, possibly due to the impact of urban land. The comparison of the results showed that the value of $R^2$ was 0.8227 and the RMSE was 8.8181, which depicted the effectiveness of the proposed multi-dimensional data-based method for estimating city-level carbon emissions. It could be utilized for the subsequent spatial analysis.

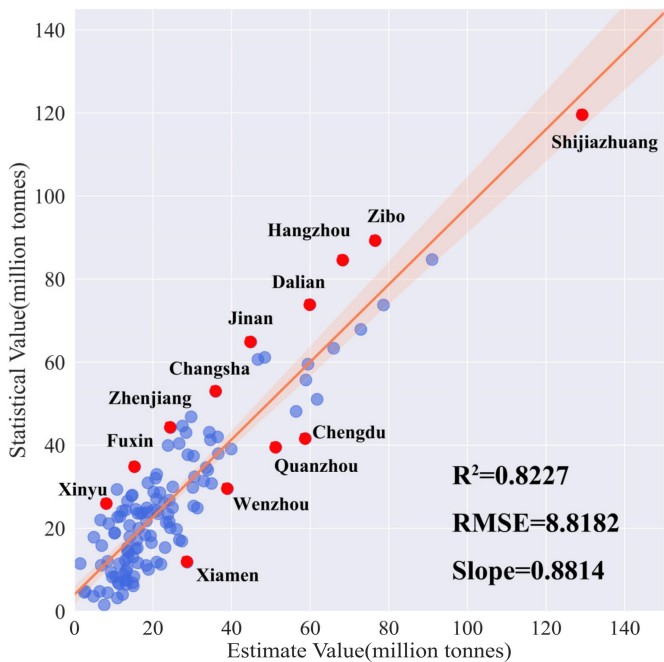

**Figure 6.** Spatial distribution of city-level carbon emissions from 2001 to 2019.

**Figure 7.** Comparing city-level results of estimation with statistical yearbook statistics.

**Table 5.** Data on the cities.

| City | AVSI | AVTI | LU | LST | NTL | TCO2 | RCO2 | CCO2 | DCO2 | PCO2 | ESGDP |
|------|------|------|------|------|------|------|------|------|------|------|------|
| Shijiazhuang | 1653.92 | 1377.75 | 1551.00 | 20.22 | 158809.21 | 119.54 | 60.06 | 10.76 | 3.32 | 25.86 | 1.49 |
| Quanzhou | 2145.03 | 1287.55 | 1409.00 | 23.66 | 140489.69 | 39.52 | 15.11 | 2.14 | 4.34 | - | 0.78 |
| Wenzhou | 1535.12 | 1297.66 | 717.00 | 20.96 | 109532.03 | 29.54 | 22.86 | 0.06 | 2.47 | 0.13 | 0.59 |
| Xiamen | 1026.86 | 1003.88 | 443.00 | 25.23 | 47943.50 | 11.85 | 6.82 | - | 1.61 | - | 0.57 |
| Chengdu | 2480.90 | 2785.30 | 1303.00 | 19.79 | 173981.74 | 41.63 | 11.58 | 3.21 | 5.14 | 5.96 | 0.72 |
| Zibo | 1766.57 | 994.88 | 885.00 | 19.79 | 91561.24 | 89.20 | 42.80 | 7.10 | 4.10 | 11.10 | 1.62 |
| Hangzhou | 2844.47 | 2893.39 | 1409.00 | 20.16 | 177833.43 | 84.56 | 54.88 | 4.02 | 4.39 | 10.21 | 0.68 |
| Dalian | 2645.50 | 2167.50 | 741.00 | 15.23 | 117448.16 | 73.86 | 34.93 | 0.77 | 9.47 | 6.43 | 0.87 |
| Jinan | 1637.45 | 2058.18 | 503.00 | 19.38 | 121026.15 | 64.88 | 20.28 | 13.04 | 4.76 | 4.78 | 1.00 |
| Changsha | 2437.03 | 1908.02 | 498.00 | 22.12 | 89807.55 | 52.97 | 12.30 | 15.56 | 2.74 | 10.32 | 0.83 |
| Zhenjiang | 1124.52 | 750.54 | 395.00 | 20.27 | 85290.35 | 44.24 | 33.31 | 1.81 | 1.01 | 5.47 | 0.80 |
| Fuxin | 148.70 | 119.70 | 171.00 | 15.93 | 33823.06 | 34.82 | 30.56 | 0.68 | 0.69 | 0.69 | - |
| Xinyu | 403.36 | 189.98 | 88.00 | 23.20 | 17229.25 | 29.40 | 8.00 | 10.60 | 0.80 | 1.20 | 2.69 |

Note: AVSI is the added value secondary industry (the unit is CNY 100 million), AVTI is the added value tertiary industry (the unit is CNY 100 million), LU is the urban land area (the unit is square kilometers), LST is the land surface temperature (the unit is Celsius), NTL is the sum of DN values in each city, TCO2 is the statistical total $CO_2$ (the unit is 1 million tonnes), RCO2 is the statistical raw coal $CO_2$ (the unit is 1 million tonnes), CCO2 is the statistical coke $CO_2$ (the unit is 1 million tonnes), DCO2 is the statistical diesel oil $CO_2$, PCO2 is the statistical process $CO_2$ (the unit is 1 million tonnes), and ESGDP is the energy consumption per unit of GDP (the unit is tons of standard coal/CNY 10,000).

### 3.2. Spatiotemporal Characteristics of Carbon Emissions from 2001–2019

#### 3.2.1. Linear Trend Analysis

An analysis of the change in slope of city-level carbon emissions can directly reflect the spatiotemporal trends of the region. Changes in the slope of China's city-level carbon emissions in the three time periods of 2001–2006, 2006–2012, and 2012–2019 are shown in Figure 8. During the two periods of 2001–2006 and 2006–2012, most cities were in a stage of slow growth in carbon emissions, although a few cities were in a stage of rapid growth, primarily located in the eastern coastal city clusters along the Bohai Bay rim, and in midwestern regions such as Chongqing, Wuhan, Chengdu, and Hulun Buir. It should be noted that from 2012 to 2019, nearly one-third of China's city carbon emissions were in the emission-reduction stage. This was related to the energy conservation and emission-reduction policies during the "12th Five-Year Plan" period (2011–2015) [56], whose specific goals provided strong policy constraints for eliminating outdated production capacity and promoting the upgrading of polluting enterprises. On the other hand, it showed a great relationship with China's air pollution prevention and control actions implemented in 2013, which had a primary goal to control total coal consumption and use clean energy such as natural gas and coalbed methane [57]. Changes in carbon emissions indicated that relevant departments across China were following "green" and "low-carbon" emission-reduction policies. Nevertheless, due to the continuous impact of industrial transformation and economic development, achieving such goals will require a long period of time.

#### 3.2.2. Standard Deviational Ellipse

Based on the SDE analysis of China's city-level carbon emissions, Figure 9 demonstrates the spatiotemporal characteristics of China's energy carbon emissions. From 2001 to 2019, the SDE centers of the national energy consumption carbon emissions were all distributed in Henan Province, where the pattern of the ellipse centers changed significantly. From 2001 to 2006, the center of the ellipse moved to the south by 26.99 km, which was ascribed to the large number of carbon emissions generated by the deep development of the urban agglomeration in the Yangtze River Delta and the Pearl River Delta, which caused the center of the ellipse to move to the south. From 2006 to 2019, the ellipse center moved rapidly to the northwest by 93.49 km, with an average annual movement of 6.68 km. Finally, the center of the ellipse fell in Zhengzhou. This is mainly related to the "Eleventh Five-Year" plan for the development of the western region passed by China in 2006, in which the target

was to establish key coal power bases in northern Shaanxi, western Inner Mongolia, and other regions, as well as oil and natural gas production bases in Xinjiang, Shaanxi, and Gansu. Even though these heavy industries brought rapid economic development, they also caused massive carbon emissions.

The long axis of the SDE generally decreased, from 2214.46 km in 2001 to 2147.81 km in 2019. On the other hand, the short axis increased, from 1594.15 km in 2001 to 2009.28 km in 2019 (Figure 10). The oblateness of the ellipse was decreasing, the long axis here was in the northeast–southwest direction, and the short axis was in the northwest-southeast direction. The directionality of city-level carbon emissions is constantly weakening, and the degree of dispersion is increasing. Combining the interannual changes of the long axis and short axis of the SDE, the impact of carbon emissions in the northeast-southwest region on China's carbon emissions gradually decreased, while the impact of carbon emissions in the northwest–southeast region on China's carbon emissions gradually increased. It was found that the direction of China's city carbon emissions in 2001 was the strongest, while that of city carbon emissions in 2019 was the weakest. The SDE difference azimuth could identify the direction of change in the carbon emission pattern, which had an obvious downward trend from 2001 to 2019. Combined with the elliptical center of gravity transfer trajectory, we concluded that the northwest direction had an increasing influence on changing the carbon emission pattern of China's energy consumption.

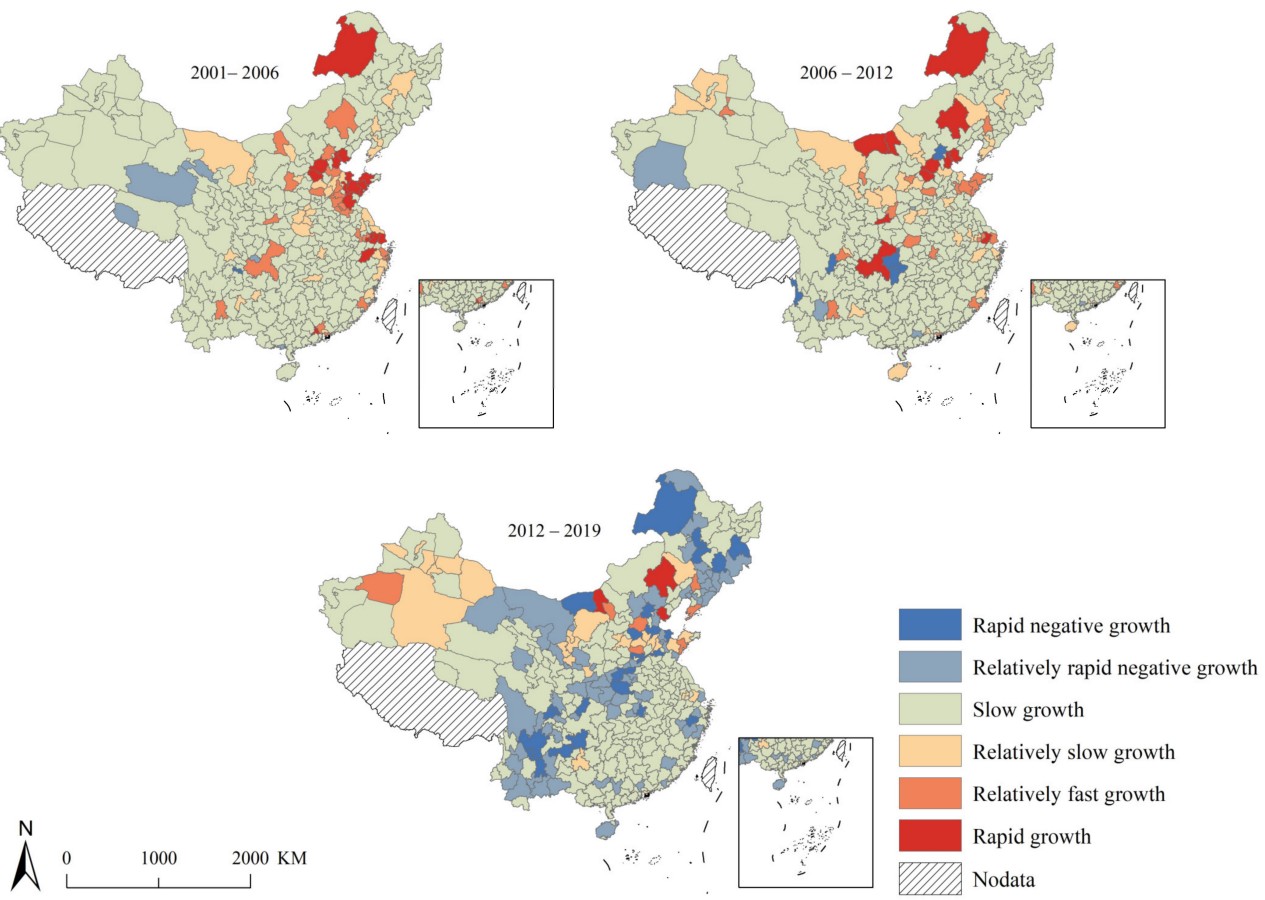

**Figure 8.** The slope of carbon emission change at the city level.

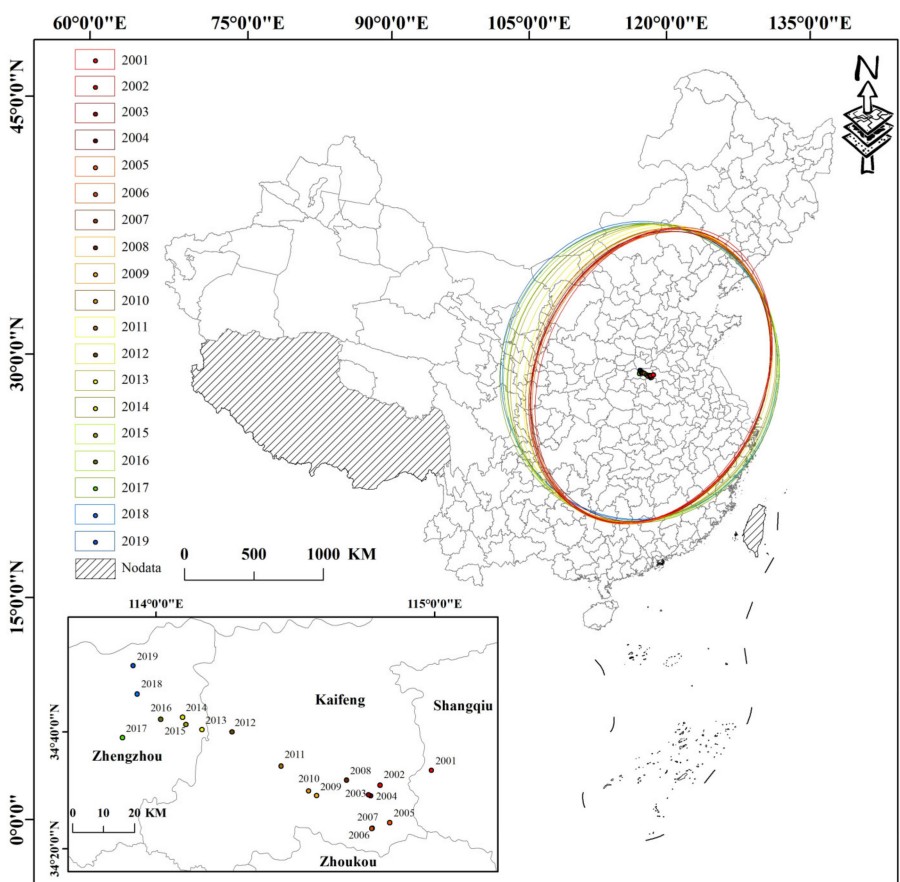

**Figure 9.** Ellipse analysis of China's carbon emission standard deviation from 2001 to 2019.

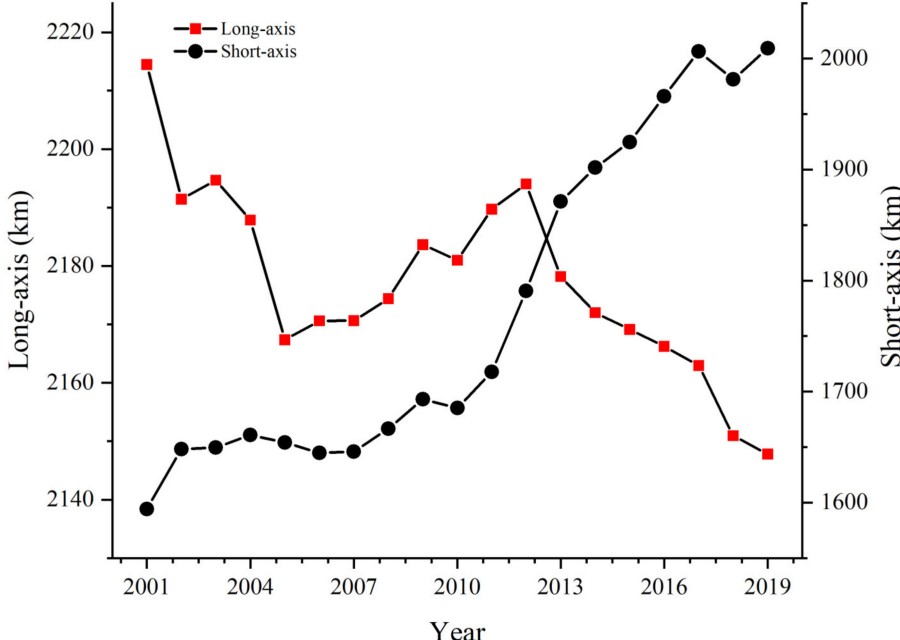

**Figure 10.** SDE long-axis and short-axis changes.

### 3.3. Discussion

This study utilized multi-dimensional data as proxy variables to estimate the carbon emissions at the city level in China from 2001 to 2019. This work confirmed a significant

regression relationship between the multi-dimensional data and CES data. Compared with the studies by Shi et al. [24] and Lv et al. [26], this study overcame the need only to consider the NTL to estimate carbon emissions, and the city-level evaluation results showed that this method was efficient and effective. However, this study still had the following limitations.

Firstly, various statistical data adopted in this study may have contained inconsistent calibers or artificial statistical errors, which could have affected the accuracy of the model training, and may ultimately have affected the results of the city-level carbon emissions estimation and analysis. For example, the method only used the total consumption of various types of fossil energy multiplied by their respective average low-level calorific value and carbon emission coefficient, but it did not consider whether these values and coefficients of each region were consistent. Moreover, there were some statistical errors and omissions in the statistical data of the AVSI data at the county level, and the estimation accuracy was difficult to verify. This issue was similar to that encountered by Zhao et al. [58], so we did not estimate the carbon emissions at the county level in China for an extended period of time.

Secondly, the radiometric resolution of the DMSP-OLS storage device utilized in the study was 6 bits; consequently, the DN range of the image was only 0 to 63. Previous studies also encountered this problem [24,26,32]. Although the mutual correction method was utilized in the NTL dataset processing to deal with the problem of interannual inconsistency, there were still issues with saturation and the blooming effect, leading to errors in carbon emissions estimation.

Thirdly, this study considered LST, AVSI, NTL, and LU data as indicators for carbon emissions estimation model, but in fact, other factors can be responsible for carbon emissions.

Furthermore, the results for spatiotemporal variations of city-level emissions showed that most of China's cities with high carbon emissions were situated in the Bohai Rim and the eastern coastal urban agglomeration, which was consistent with the findings of Sun et al. [22]. In summary, based on multi-dimensional data and machine learning, a novel and effective method was provided to simulate the spatiotemporal evolution of Chinese cities. Due to the lack of statistical data, county-level carbon emissions data were challenging to estimate and verify.

## 4. Conclusions

This study established a correlation between multi-dimensional data and carbon emission statistics through the DNNE model to analyze the spatiotemporal changes in carbon emissions in Chinese cities from the period of 2001 to 2019. The conclusions are as follows.

Compared to previous studies, this work used machine learning and considered multi-dimensional data to estimate carbon emissions at the city level in China, which was a novel and effective method. Its advantage lay in considering the significant regression relationship between the multi-dimensional data and carbon emissions data. Consequently, not only city-level carbon emissions could be estimated quickly, but the results were also highly accurate. The evaluation results showed that the method proposed in this study was suitable for city-level carbon emissions estimation, and can be considered a way to address the lack of statistical data. In addition, the spatiotemporal variations of city-level emissions were analyzed in this study. Results of the linear trend analysis showed that there were differences in the growth of carbon emissions in different regions of China. Most of China's cities with high carbon emissions were situated in the Bohai Rim, the eastern coastal urban agglomeration, and midwestern regions such as Chongqing, Wuhan, and Chengdu. Nearly one-third of the cities in China have reduced carbon emissions since 2012. The SDE analysis indicated that the azimuth angle of China's carbon emission SDE showed a decreasing trend, and the center continues to move to the northwest. Although there were some limitations, this study can also provide an accurate way of estimating

carbon emissions by considering multi-dimensional data, which can also serve as a guide for energy saving and emission-reduction measures.

**Author Contributions:** Conceptualization, J.M.; Formal analysis, X.L., H.C., and F.S.; Methodology, X.L. and J.M.; Supervision, Z.L.; Validation, X.L.; Visualization, X.L. and J.M.; Writing—original draft, X.L.; Writing—review and editing, X.L. and S.A. All authors have read and agreed to the published version of the manuscript.

**Funding:** This study was supported by the National Natural Science Foundation of China (Grant No. 41671352) and the Natural Science Foundation of Anhui Province, China (No 1908085QF279).

**Data Availability Statement:** Not applicable.

**Conflicts of Interest:** The authors declare no conflict of interest.

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
