# Peer review of "Carbon Emissions Estimation and Spatiotemporal Analysis of China at City Level Based on Multi-Dimensional Data and Machine Learning"

_remotesensing, doi:10.3390/rs14133014_

Round 1

Reviewer 1 Report

Dear all,

Lin et al. tried to estimate Carbon emission on the urban scale by using the relationships found on a regional scale. I am not fully experienced with the above topic and I will discuss my points from the point of view of applying machine learning methods to air pollutants. In my opinion, first, the following points need to be better discussed in the next version of MS: 1- A limited number of influential factors are discussed for the development of the model. I suggest that authors expand this section by using other potential factors such as population and then identify the important factors by applying feature reduction methods. 2- Nighttime light data, which are of great importance in model development, have been extrapolated. The accuracy of this data is still debatable. The extrapolation certainly causes uncertainties. Can you suggest a method to quantify uncertainties? 3- Your observational data, based on my understanding, are data reported by the government or other institutions. I suggest comparing this data with other sources such as satellite products. Please also explicitly name your observations in figures. I think you called them here statistical values? right?

Other points:

lines 89-93: do not discuss results here.

line 128: the caption of tables and figures is inserted in the text.

line 131: not meaningful to me. Of course, any data can be directly fed into MLA but will be more informative if they better explain the predictant. 

Figure 7. Please prove a clearer caption. I suggest putting observations in the y-axis

line 373. Please provide a clearer explanation regarding the relationship between changes in SDE (long and short axes) and emission.

 Best of luck

Author Response

Dear Reviewer 1,

We are thankful to the reviewer 1 for providing valuable comments and suggestions to our submitted manuscript entitled “Carbon Emissions Estimation and Spatiotemporal Analysis of China at City-level Based on Multi-dimensional Data and Machine Learning”. These precious comments are important for improvement of our manuscript and a better guidance for this research. We have thoughtfully considered these suggestions and have tried our best to revise and improve the manuscript. Please refer to the re-uploaded manuscript and the attachment for details of the changes.

Thanks again for your valuable suggestions and comments on this manuscript. We wish you more fruitful achievements in research, and more happiness in life.

Best regards

Yours sincerely

Jinji Ma

Reviewer 2 Report

- line 96 - remove the dot between "study" and "shows"

- lines 121-123 -  you already mentioned the meaning of all these abbreviations in the Abstract (lines 19, 20) and in the Introduction (lines 55, 80,81) - no need to duplicate this

 - line 129 - duplicate word "Table"

- line 130 - remove the dot between "list" and "outlines" and make space between these words

- line 175 - remove the dot between "2019" and "shows"

- line 189 - make space between "7.5:2.5" and "(Devide..."

- no need to duplicate explanations of abbreviations threw the paper

- line 257 - type error - duplicate words "Table 2" and remove the dot between "classification" and "is"

- Figures 3 & 8 and lines 339-353 ==> can you explain why the province Xinjiang has continuous growth of carbon emissions? Why provinces Inner Mongoria, Henan, Zhejiang and Heilongjiang have its highest peak around 2010 and then values of carbon emissions are dropping? And why provinces Hebei, Shandong, Liaoning and Jiangsu have this rapid growth of carbon emissions in 2018/2019?

Author Response

Dear Reviewer 2,

We are thankful to the reviewer 2 for providing valuable comments and suggestions to our submitted manuscript entitled “Carbon Emissions Estimation and Spatiotemporal Analysis of China at City-level Based on Multi-dimensional Data and Machine Learning”. These precious comments are important for improvement of our manuscript and a better guidance for this research. We have thoughtfully considered these suggestions and have tried our best to revise and improve the manuscript. Please refer to the re-uploaded manuscript and the attachment for details of the changes.

Thanks again for your valuable suggestions and comments on this manuscript. We wish you more fruitful achievements in research, and more happiness in life.

Best regards

Yours sincerely

Jinji Ma

Reviewer 3 Report

The manuscript describes a deep neutral network ensemble (DNNE) model, which analyzes the nonlinear relationship between carbon emission statistics and multi-dimensional data, including nighttime light data, land use data, land surface temperature data, and added value secondary industry data in China from 2001 to 2019. The model was also compared with two other methods: multiple linear regression and random forest. The authors concluded that DNNE works the best in terms of R2 and RMSE to estimate city-level carbon emission. On top of the model, the authors also performed linear trend analysis and standard deviational ellipse analysis. They concluded that the spatiotemporal changes in the carbon emission at the city-level are in line with the development of Chinese economy.

The manuscript was well written. The research questions and steps were generally easy to follow. The authors also managed to draw a reasonable conclusion based on the available results. However, some specifications of method/data used remained unclear to me. I suggest publication after considering the concerns I have as below.

General comments:

  1. Figure 1: This 3-D figure does not show all the column bars as some are obscured. It would look better visually to use a color scale instead of 3-D representation.
  2. The data sets listed in Table 1 is a bit confusing. I guess not all the data of the whole time interval was used. I suggest to list only the data used in this study to avoid confusion. For example, I guess data between 1992 to 2001 were not used in the study.
  3. Table 2: Do the authors have a reference for this classification? Was it used in other papers as well? And please make the minus signs and dash signs distinguishable in the table.
  4. Why did the authors choose the three methods? For neural networks, there are also some time-dependent methods, like long short-term memory (LSTM) method? Have they considered these before?
  5. Table 3: Now the comparison of the three methods are only in terms of R2 and RMSE. Have the authors considered the computational time and resources as well? Random forest and neural networks are more complicated than linear regression, so it is quite obvious that they have higher accuracy.
  6. Figure 4: Can the authors explain more about the increasing trend of the DN numbers. Does it mean that there are more and more data each year and there is a need to calibrate the model separately each year?
  7. Does Figure 5 show the results from training set or testing set? The numbers are different from the ones in Table 3. Please specify.
  8. Section 3.1.1: The description of R2 and RMSE could have been moved to the method section.
  9. The captions of all figures and tables were included in the main text. Please check again before later version.
  10. Figure 6: Did the authors show the total city-level carbon emissions or the carbon emissions per area? If it was a total emission for that city, the larger the city, the higher the emissions. This would create a bias to show the data in color on a map.
  11. Figure 7: Why did the authors choose to highlight these cities?
  12. Ln 163: Is taking the Sicily area for correction a normal procedure? Any reference for this step?
  13. Ln 183: Are carbon emissions mainly due to industrial consumption in China? How about agricultural activities and traffic activities? Any reference for this statement?
  14. Ln 189: 7.5:2.5-->3:1
  15. Ln 241: Took--> It took
  16. Ln 277: ‘evaluate of the result’-->’evaluate the result’
  17. Ln 422: fellow-->follows

Author Response

Dear Reviewer 3,

We are thankful to the reviewer 3 for providing valuable comments and suggestions to our submitted manuscript entitled “Carbon Emissions Estimation and Spatiotemporal Analysis of China at City-level Based on Multi-dimensional Data and Machine Learning”. These precious comments are important for improvement of our manuscript and a better guidance for this research. We have thoughtfully considered these suggestions and have tried our best to revise and improve the manuscript. Please refer to the re-uploaded manuscript and the attachment for details of the changes.

Thanks again for your valuable suggestions and comments on this manuscript. We wish you more fruitful achievements in research, and more happiness in life.

Best regards

Yours sincerely

Jinji Ma

Round 2

Reviewer 1 Report

no comment.